# Experiences of people with dual sensory loss in various areas of life: A qualitative study

E. Veenman [1,2]*, A. A. J. Roelofs[3], M. L. Stolwijk[1,2], A. M. Bootsma[3], R. M. A. van Nispen[1,2]

**1** Amsterdam UMC Location Vrije Universiteit Amsterdam, Ophthalmology, Amsterdam, The Netherlands, **2** Amsterdam Public Health, Quality of Care, Amsterdam, The Netherlands, **3** Royal Dutch Visio – Het Loo Erf, Apeldoorn, The Netherlands

\* e.veenman1@amsterdamumc.nl

**Data Availability Statement:** The data are not publicly available due to their containing information that could compromise the privacy of research participants. A summary of the data and

## Abstract

Individuals with dual sensory loss (DSL) appear to have limited ability to compensate for their visual impairment with residual hearing, or for their hearing impairment with residual vision, resulting in challenges in various areas of life. The aim of this qualitative study was to explore the diverse experiences facing individuals with DSL as well as to determine how they experience sensory compensation. Semi-structured interviews were carried out in twenty adults with DSL (13 females and 7 males, mean age 47 years). The causes of DSL severity varied amongst participants. Sensory compensation and experiences in regards to access to information, mobility, communication and fatigue were discussed. Interviews were audio recorded and transcribed verbatim. Framework analysis was used to summarize and interpret the data. In relation to access to information, our results show that, despite various challenges, the use of assistive technology such as voice command functions, enabled participants to operate effectively. Regarding mobility, most participants were capable of finding their way in familiar environments. However, if the setting was unfamiliar, assistance from others or reliance on navigation applications was necessary. Participants experienced little issues with having conversations in quiet settings, however, crowded settings were considered very difficult. The final results showed that most participants suffered from fatigue. Carefully considering which daily activities were feasible and having a daily routine helped to cope with fatigue. This study revealed the experiences of individuals with DSL in important areas of life. The results suggest that, even though many challenges are experienced, individuals with DSL are resourceful in finding compensation strategies. However, capturing participants' sensory compensation experiences was challenging.

## Introduction

Individuals with dual sensory loss (DSL) have visual and auditory impairments. DSL can be regarded as a spectrum ranging from total vision and/or hearing loss to having some residual vision and/or hearing [1], which means that the severity of the impairments can vary within this population. The prevalence of DSL is estimated to be between 0.2 and 2%, and increases

the code book are available from https://doi.org/10.17026/dans-zpf-57wx. Interview guide is included as supplementary material. Additional sharing conditions will be considered while respecting the type of informed consent provided by the subjects and discussed with the support of RDM experts where appropriate. Data inquiries can be sent to the Medical Ethics Review Board of Amsterdam University Medical Centers - location VU University Medical Center (metc@amsterdamumc.nl, please mention the reference number 2020.467).

**Funding:** This study was funded by the Visio Foundation (https://visiofoundation.org). The funders had no role in study design, data collection and analysis, decision to publish, or preparation of the manuscript.

**Competing interests:** The authors have declared that no competing interests exist.

with age [2, 3]. Due to global demographic aging, the number of individuals with sensory loss is expected to rise [4], making DSL an increasingly relevant topic to address in health research and policy.

Previous research has shown that visually impaired individuals compensate for their vision loss by using their hearing [5]. Likewise, individuals with hearing loss compensate with their vision [6]. Neurological studies have revealed that the brain areas associated with vision in sighted individuals are used for auditory processing in visually impaired individuals [7], and vice versa in individuals with hearing loss [8]. However, having both visual and auditory impairments makes the compensatory function of both senses less obvious, and seems to result in difficulties that may go beyond the consequences of having a single sensory impairment; a principle that can be exemplified by 1+1 = 3 [1, 9].

Various aspects of life can be affected in individuals with DSL. Deafblind International and both the Nordic and Dutch definitions of deafblindness mention access to information, mobility and communication as areas of life that may be problematic for individuals with DSL [10–12]. Previous studies offer support for this statement. Regarding access to information, one study identified the problems that individuals with DSL experience while using technology [13]; for example, when watching television it is difficult to understand speech whilst background music is simultaneously playing during the show. Additionally, a review of DSL literature confirms that problems in the areas of communication and mobility are often experienced by individuals with DSL [14]. For example, it was found that difficulties in identifying facial expressions negatively influenced speech understanding. Moreover, challenges with regard to mobility, such as finding one's way and using public transportation, were identified. The Dutch functional definition of deafblindness identifies fatigue as another important life area that can cause challenges for individuals with DSL [12]. For example, Wahlqvist et al. [15] investigated the mental health of individuals with Usher syndrome type 1. This is a rare (3–6 out of 100,000 [16]), genetic disease that is associated with both vision and hearing loss. Many types exist, and type 1 is characterised by congenital deafness and progressive vision loss. The results of this study showed that more than sixty percent of the participants had fatigue-related complaints. Several studies have shown that these areas affect quality of life of other populations as well, such as elderly or individuals with specific diseases [17–20].

Although some studies give valuable insights into the impact of DSL, problems of individuals with DSL are still not fully understood. Previous qualitative research on DSL did focus on one or more of the aforementioned areas (i.e. access to information, mobility, communication, and fatigue) [19, 21], but were often aimed at specific groups (e.g. older adults [19] or individuals with Usher syndrome [21]). A smaller number of qualitative studies have focused on compensation strategies of individuals with DSL [22], however, to our knowledge, no qualitative studies have been conducted in which the role of vision and hearing as compensatory senses in DSL was addressed. This information might allow individuals with DSL to benefit from strategies (e.g. learn how to use their remaining vision and/or hearing as a compensation strategy) that lead to improved access to information, mobility, communication and to less fatigue. Consequently, this information may also be of importance to healthcare professionals working with the target population as this might help them improve the care they provide. The objective of this study was to explore experiences of individuals with differing causes and levels of DSL regarding their access to information, mobility, communication, and fatigue. In addition, this study aimed to clarify how these individuals utilize their visual and auditory senses within these areas.

## Methods

### Study design & participants

This qualitative, explorative study was conducted between January and July 2021 in the Netherlands. The COnsolidated criteria for REporting Qualitative research (COREQ) checklist [23] was used to verify that the study was conducted appropriately.

Comprehensive interviews were undertaken with Dutch-speaking adults, who exhibited diverse degrees of impairment in both vision and hearing. To define vision and hearing loss, the following criteria were used: a best-corrected visual acuity of < 0.3 in the better eye, and/or a visual field of ≤ 30 degrees around the central fixation point [24], and a hearing threshold of >25 dB [25]. In a number of cases, we slightly deviated from these norms. As we were interested in as many different combinations of impairment severity as possible, participants who were close to these norms were included as well. Exclusion criteria were non-congenital brain damage or impaired cognitive functioning.

Two channels were used to purposively recruit a sample of participants. First of all, health professionals employed by an inpatient low vision service center in the Netherlands (Royal Dutch Visio) were asked to recruit potential participants. In addition to offering support (e.g. computer training and mobility training) to visually impaired individuals, Royal Dutch Visio also has clients with DSL. Secondly, a call for participation was posted on several of Royal Dutch Visio's social media channels (Facebook, Instagram, Twitter and LinkedIn). If individuals were interested to participate, they contacted (i.e. telephoned or emailed) the executive researcher, who provided further information, and sent them an information letter and informed consent form (either a physical or digital copy, depending on the participants' preferences). After receiving the signed informed consent form, an appointment was made to conduct the interview. In total, 20 interviews were scheduled which we anticipated to be the number required to achieve data saturation.

### Interviews

Semi-structured, in-depth interviews were conducted by the first author (MSc. Cognitive psychology). An interview guide was developed in cooperation with healthcare professionals (e.g. computer trainer, mobility trainer) who were familiar with clients with DSL and are experts in the areas of interest. The interview guide consisted of an introduction and open-ended questions about sociodemographic and medical information, and the four areas of interest mentioned earlier (see S1 Appendix). More specifically, the guide consisted of questions regarding which challenges or potential favourable factors individuals experienced in the four areas, how they coped with these challenges, and how they used their vision and hearing while performing certain activities. The interview was pilot tested under supervision and with a volunteer with DSL. Participants received a €20 gift voucher after finishing the interview. Interviews were audio-recorded and transcribed verbatim. With participants' permission, their medical records were requested to collect information about their diagnosis, visual acuity, visual field and hearing threshold. Most of the retrieved information was less than three years old.

### Data analysis

The data were analysed by means of framework analysis [26] using ATLAS.ti 9 software. To get familiar with the data (step 1), two researchers (EV and MS) read and discussed two interview transcripts. As a second step, the main themes were determined, by combining both a priori knowledge retrieved from the literature and the information gathered from familiarization. This resulted in four main themes which corresponded to the topics of the interview guide (i.e.

access to information, mobility, communication and fatigue). Next, two interview transcripts were independently open coded, and subthemes were identified by means of face-to-face discussion. The main themes and subthemes were combined in a codebook, which was used to code two new transcripts. The process of coding and adapting the code book was repeated twice. Subsequently, a single researcher (EV) coded the remaining transcripts using the finalized codebook (step 3). During this process, no additional changes were made to the codebook, suggesting that data saturation was reached [27]. Finally, the coded data was summarized for each theme and subtheme (step 4) and interpreted (step 5) by means of comparing the data across participants and identifying patterns in the data.

## Ethics

The Medical Ethics Review Board of Amsterdam University Medical Centers—location VU University Medical Center—determined that the present study is not subject to the Medical Research Involving Human Subjects Act (Dutch law). The study was conducted in accordance with the principles of the Declaration of Helsinki.

## Results

### Participant characteristics

Twenty adults with DSL participated in this study. Participant characteristics can be found in Table 1. The mean age of the participants was 47.1 years and ranged from 20 to 88 years. Their DSL was caused by various conditions. Visual acuity, visual field loss, hearing loss and the presence of asymmetrical hearing differed among the participants. Binaural hearing aids were utilized by 70% of the participants. Fig 1 shows visual acuity, visual field and hearing loss information of all 20 participants included in this study.

Due to COVID-19 restrictions, 10 participants preferred to be interviewed via an online video call or telephone. The other interviews were conducted at participants' homes. Most participants were capable of having a spoken conversation. However, two participants used sign language as their main means of communication, thus, a speech-to-text interpreter was present at the interview. The interpreter transformed the spoken words of the interviewer to text, which participants could read, in a large type font, on a computer screen. These two participants provided their answers in spoken form. Interviews lasted between 37 and 97 minutes, with an average of 65 minutes.

Four main themes were derived from the data: access to information, mobility, communication and fatigue. Three of these main themes were divided in subthemes; for access to information, these were smartphone, computer and tablet usage and watching television. Additionally, mobility and communication were divided into subthemes of walking, the use of public transportation, face to face conversations and conversations by telephone, respectively.

### Access to information

Regarding access to information, participants discussed their experiences related to smartphone, computer and tablet usage, and watching television.

**Smartphone.** Most participants owned a smartphone, and used it regularly. In general, they were well able to use the applications and look up information on the internet. Participants with peripheral visual field defects preferred using a smartphone over a computer or tablet: *"[When using] the smartphone, and of course that's because it is small, it makes me see it better. The smaller something is, the better we [individuals with peripheral visual field defects] can see it (female, aged 56)."* Most participants used voice control, a dark mode (i.e. black

**Table 1. Participant characteristics (N = 20).**

| Participant characteristics | Categories | N (%) | Mean (SD) [range] |
|---|---|---|---|
| Age | | 20 (100%) | 47.1 (17.1) [20–88] |
| Gender | Female | 13 (65%) | |
| | Male | 7 (35%) | |
| Living situation | Living with partner or family | 12 (60%) | |
| | Living alone | 8 (40%) | |
| Diagnosis[a] | Disorders of sclera, cornea, iris, ciliary body and lens | 3 (15%) | |
| | Disorders of choroid and retina [incl. Usher syndrome] | 11 (55%) [9 (45%)] | |
| | Glaucoma | 3 (15%) | |
| | Disorders of optic nerve and visual pathways | 3 (15%) | |
| | Other | 3 (15%) | |
| Visual acuity of the best eye | Normal vision ($\geq 0.8$) | 0 (0%) | |
| | Near-normal vision (0.3–0.8) | 10 (50%) | |
| | Moderate low vision (0.1–0.3) | 4 (20%) | |
| | Severe low vision (0.05–0.1) | 1 (5%) | |
| | Profound low vision (0.01–0.05) | 2 (10%) | |
| | Total blindness ($<0.01$) | 3 (15%) | |
| Visual field | Peripheral loss | 17 (85%) | |
| | Central loss (5˚-10˚) | 15 (75%) | |
| | Central loss (0˚-5˚) | 9 (45%) | |
| Use of visual aids | Cane | 18 (90%) | |
| | Magnifier | 5 (25%) | |
| | Assistive computer software | 5 (25%) | |
| | Guide dog | 4 (20%) | |
| | Sunglasses/cap/hat | 4 (20%) | |
| | Braille display | 2 (10%) | |
| | Assistive device for television | 2 (10%) | |
| Hearing loss of the best ear | 0–25 dB | 3 (15%) | |
| | 26–40 dB | 4 (20%) | |
| | 41–60 dB | 4 (20%) | |
| | 61–80 dB | 4 (20%) | |
| | $\geq 81$ dB | 5 (25%) | |
| Asymmetrical hearing | Yes | 8 (40%) | |
| | No | 9 (45%) | |
| | Unknown | 3 (15%) | |
| Use of auditory aids | Hearing aid monaural | 2 (10%) | |
| | Hearing aid binaural | 14 (70%) | |
| | Cochlear implant monaural | 2 (10%) | |
| | Cochlear implant binaural | 2 (10%) | |
| | Streamer | 4 (20%) | |
| | Remote microphone | 1 (5%) | |

[a] Participants could have more than one diagnosis

background and white text), or magnified the font to facilitate smartphone use. However, some difficulties were associated with this as well. For example, some applications did not work properly with magnified text, and the absence of a dark mode in some applications prevented individuals who are sensitive to light from using the application. Additionally, mistakes

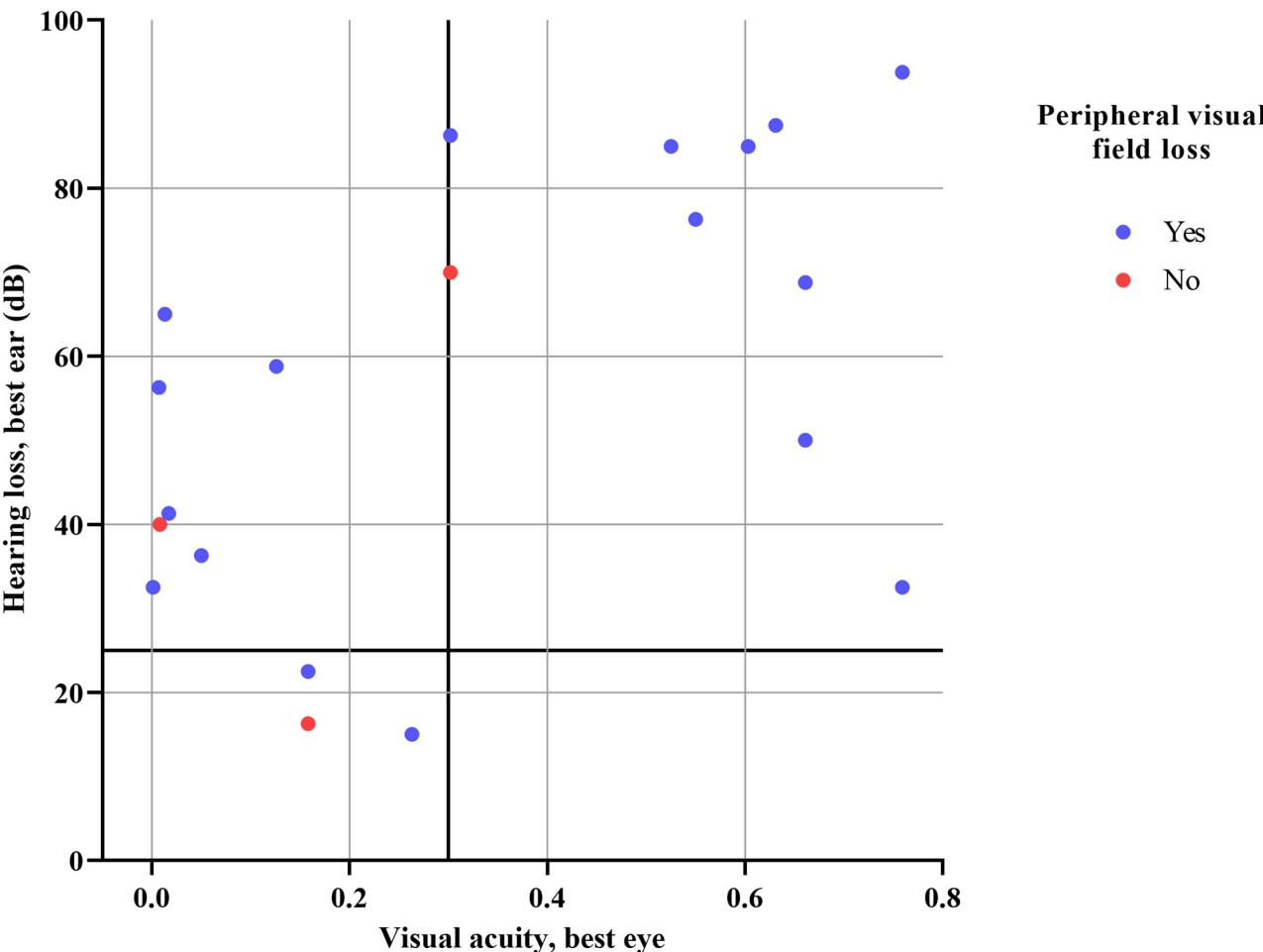

**Fig 1. Participants' visual acuity, peripheral visual field loss and hearing loss.** The vertical and horizontal lines indicate the norms for having a visual and auditory impairment, respectively (i.e. visual acuity = 0.3 and hearing loss = 25 dB).

were regularly made with the dictation function; spoken words were not always properly transformed into text, and these mistakes could not be detected easily because of the visual impairment. Controlling the smartphone so that a web page is being read aloud came with certain difficulties as well: *"I have to select what I want to have read aloud. So I cannot say: 'read this page aloud'. (. . .). With an article, I might be selecting the text six times, because I have to select the first piece of text, then there is an advertisement, then I have to select the next piece of text, scroll again, select again. (. . .) That makes me think: 'well, never mind then' (male, aged 31)."* Another challenge that many participants referred to, was that they had to put in a lot of effort to be able to understand speech when using the screen reader function of their smartphones. However, the possibility of directly connecting their smartphones to their hearing aids or cochlear implants was regarded as useful in this case.

**Computer/Tablet.** Computers and tablets were also regularly used by most participants; for example, to visit internet pages and to send e-mails. Participants with decreased visual acuity preferred using a computer or tablet over a smartphone: *"There are many options on the tablet to magnify things, and you can just tick on it, and it magnifies, so that is really nice. I do almost everything on my tablet nowadays (female, aged 37)."* In contrast, individuals with peripheral visual field loss often mentioned that they were not able to get an overview over the

entire computer/tablet screen, and therefore preferred using a smartphone. Other difficulties associated with computer and tablet use were related to visual aspects, such as being unable to distinguish the icons and constantly losing the cursor on the screen. To cope with this, some participants used supportive software: *"With that program, I can easily change the font size. [. . .] Now, I need a font size of 28 or 35 to distinguish what I am writing. [. . .] The program also has an excellent screen reader (male, aged 88)."* Other participants used a dark mode or a braille display to make computer/tablet use easier. Also, some participants mentioned that they received a computer training from a low vision service; this was considered useful, because they were taught adequate skills to facilitate their computer and tablet usage.

**Television.** When watching television, participants were able to enjoy well-structured programs, like the news or certain talk shows, because these are often slowly paced and easy to follow. The faster paced types of television content, however, did cause problems: *"Well, when I am watching a movie, (. . .) when many things are happening, like in an action movie, I over-look half of it. (. . .) Or when things are happening in the dark, well, I just cannot see it. And I try to, but I notice that I get the feeling that I should not continue watching. It is so exhausting (female, aged 56)."* These difficulties caused some participants to avoid watching television at all. Similar to computer and tablet use, another frequently mentioned problem was that a large screen is difficult to oversee for individuals with peripheral visual field loss. To compensate, they used their smartphone or tablet to watch certain programs on a smaller screen. In turn, other participants were forced to sit close to the television to be able to see what was happening. Auditory impairments caused difficulties as well, especially when many people on television were talking simultaneously or when the spoken language was not the same as the participant's native language. However, directly connecting the television sound to one's hearing aids or cochlear implants was helpful. Another difficulty was not being able to easily read subtitles. Certain assistive technology could solve that problem: *"You can change the background [of the subtitles] and the size. [. . .] I am happy with it, because if there's someone with a white t-shirt [on television], I cannot read the subtitles (male, aged 52)."* Participants also made use of assistive technology that reads the subtitles aloud. Notably, some participants mentioned that they relied on either vision or hearing while watching television. One participant mentioned: *"At 8 P.M., we watch television for a while. That is, I listen to what is being said, and my wife watches (male, aged 88)."* Another participant experienced watching television differently: *"I often watch a soap series with subtitles with my daughter. [. . .] If there are no subtitles, I cannot understand anything (female, aged 41)."*

## Mobility

In the area of mobility, participants were asked how they experienced travelling on foot and by public transportation. Some findings applied to both travelling modes. Most participants indicated to find it easier to travel in familiar environments as opposed to unfamiliar environments. *"In familiar settings, I can move around by myself pretty well. [. . .] Familiar ways are fine. And 'being familiar' is key; I have to know a route well to be able to walk it by myself. I can't just go to a new place by myself (female, aged 27)."* Unfamiliar environments tended to influence the participants' confidence: *"So you are more insecure, but that is caused by dual sensory loss, because you cannot rely on either of the two senses, you know? And that's just frustrating (female, aged 42)."* A consequence of this reduced confidence was that some participants avoided going to unfamiliar places alone. Another way to deal with difficulties regarding mobility, was to seek help from low vision services: *"We go to the shopping mall or a large traffic interchange [. . .] He teaches me to listen to where everything is coming from. [. . .] Well, signalling, so using your cane when you cross the street, walking upright, [. . .] because your cane does*

*the work for you (female, aged 49).”* Another general finding was that participants' vision was negatively influenced by bad light conditions. For example, one participant mentioned: *“When it's light, there's no problem, but when it's very dark, I think it's terrible. I really don't see anything (male, aged 74).”* On the other hand, some participants also experienced problems with bright light. To deal with bad lighting situations, some participants used sunglasses, a hat, or street lights to navigate. Others, however, avoided travelling when light conditions were not sufficient.

**Walking.** Most participants were able to move around on foot, although crossing the street was considered a difficult situation. Participants had difficulties determining the location or direction of approaching cars or cyclists: *“But you have to be able to hear where the cars are coming from and if there is a road divider [. . .] and if there is a bicycle path. These are difficult things (female, aged 68).”* To make crossing the street easier, participants used a pedestrian crossing or traffic light, or asked others to walk with them. Participants took their time to make sure a traffic situation was safe. Most participants tended to use their hearing to determine whether there was any traffic approaching; hearing aids and/or a cochlear implant were considered helpful in this situation. When they detected a road user with their hearing, they used their sight to locate that road user more precisely. Other problems in traffic that were mentioned were that other road users or obstacles on the sidewalk were being overlooked at times and that electric cars and cyclists were difficult to hear. Other traffic was also difficult to hear when there was background noise: *“Some you hear, some you don't. [. . .] When it is very crowded, I cannot hear whether a car is approaching. When people are working in the garden here, and you have to cross the street there, I cannot hear it, it is disrupting (female, aged 41).”* Most participants used a cane or guide dog when walking: *“It makes you feel more secure and it makes you walk faster. I really need it; I cannot cross the street without my cane (female, aged 51).”* A cane or guide dog had other advantages as well: *“Look, people have to see that I cannot see it, because, if I didn't have a cane, they would've thought: 'Why is that woman walking so slowly?' or 'Hurry up!' [. . .] You can see that parents grab their children when they are walking in front of me. [. . .] People let you go first. Or if you're hesitating, people ask 'Can I help you?' (female, aged 51).”*

When finding their way, most participants used navigation applications (e.g. Google Maps) on their smartphone. These applications were considered helpful: *“It makes you more confident. The problem when going to a new place is that you don't recognise the buildings, and now you can just turn it on and then at least you're in the right street (female, aged 42).”* A problem some participants mentioned when finding their way, even when using an application, was to find a specific house. A way participants coped with experienced problems when finding their way was to ask other people for help: *“I haven't been to the city centre in a long time. It has completely changed. I do want to go exploring it there, but I want to go with a volunteer. Look, if I've been to a place once, I know the way (male, aged 52).”*

**Public transportation.** Most participants prepared their journey before travelling with public transportation. *“I think it is important to make a plan beforehand, to make sure I know what to look out for. [. . .] That you know where you will sit, close to the travel information screen [. . .] and to the exit. The further you go, the more obstacles you can run into (male, aged 27).”* Some participants were able to use their eye sight, which made using public transportation easier. For example, they were able to differentiate buses and could read travel information displays at train stations. Others were not able to use their eye sight for this purpose. As a solution, they used smartphone applications to retrieve travel information and to find their way in train stations. Another way to deal with this was asking others to help them or travel with them, or following other people when stepping out of the train, because this strategy would take them to the station hall. Problems were experienced when trying to understand

speech in stations, as this is usually a noisy and reverberating setting. *"I do ask others some-times, but it's difficult, because often there's a lot of noise and then I can't hear what the answer is (female, aged 52)."* Some participants also had difficulties understanding announcements in the train or bus (e.g. what the following stop was). They had a practical solution for this prob-lem: *"Or when you know beforehand that there will be six stops, and you hear a computer voice six times, then you know [when you have to get out] (male, aged 27)."*

## Communication

The area of communication encompassed having conversations face to face and over the tele-phone. A general challenge that was mentioned, was that other people do not understand the hearing impairment: *"It's the incomprehension of other people, when I say: 'I cannot understand you', they turn up the volume, and still I have to say: 'I still cannot hear you.'. And then they say 'But I am speaking up, right?' Yes, but you're screaming. I find that difficult sometimes (female, aged 56)."* Another difficulty mentioned by some participants was that it is difficult to see facial expressions or to determine by someone's voice which emotion is expressed: *"Sometimes you make a fool of yourself [. . .]. Sometimes if someone makes a joke, and then you think [. . .] that they are being serious and then a misunderstanding occurs (female, aged 52)."* The COVID-19 pandemic also negatively influenced participants' experiences with communication: wearing face masks prevented them from speech reading, and social distancing made it even more diffi-cult to understand another person. Some physical complaints were mentioned as well; as a result of bending towards someone to better understand them, some participants had a bad posture. Communication also affected participants' mental wellbeing: *"There are moments when I can't understand everything and that can be frustrating [. . .] Sometimes I feel like an out-sider. Then I think 'okay, never mind. I will find out later' (female, aged 20)."* Others mentioned that they became isolated, because of avoiding social activities.

**Face to face conversations.**   There was one predominant finding regarding communicat-ing face to face, namely that having conversations in a busy environment caused a lot of diffi-culties. *"For example, in a group where multiple people talk over each other, I cannot handle that. [. . .] You are expected to smile and enjoy the atmosphere, but I cannot follow the conversa-tion anymore (female, aged 68)."* It caused difficulties as well when there were multiple conver-sations in the same room at the same time: *"Yes, it is hard for me to focus on these conversations. Especially when the others are talking loudly, and I am talking to someone with a soft voice, I find it difficult to understand that person. [. . .] So, it is difficult for me when multiple conversations are happening at the same time (male, aged 44)."* The fact that there was often a lot of background noise, made it especially difficult to understand conversations: *"I hear the conversations that others are having, and I try to understand the conversation that I am in, but quite often there is music playing or something, so it is hard to obtain the information that I want (male, aged 31)."* Another difficult situation was when the person talking was too far away to understand what they were saying. Visual aspects caused problems in busy environ-ments as well, for example, it was considered difficult to oversee the conversation and deter-mine who was talking. Lighting conditions mattered as well: *"If we go somewhere and it is very dim in a living room, this especially bothers me in the winter, if I visit someone who has ambient lighting, that's a problem for me, because I cannot rely on my hearing, so I have to properly see the person who is talking (female, aged 56)."* The specific difficulties associated with dual sen-sory loss were mentioned as well: *"I notice that people that I know who have a hearing impairment, they can speech read, and that is not applicable to me, so in a busy environment that's very difficult for me (female, aged 27)."* Although most participants owned a hearing aid, some participants did not consider them to be very helpful in these situations: *"The hearing aid*

*amplifies everything, so you hear a lot of noise (female, aged 27).*" Others did benefit from their hearing aids, for example because the aid took the direction of their gaze into account. Participants also benefited from other aids: *"If my husband and I go out for dinner, he has a microphone [which is connected to the hearing aid and cochlear implant], then he talks in the microphone [. . .]. That's very nice, because I can understand him much better (female, aged 52)."* Other compensation strategies in busy environments were explaining the impairments to others, asking others to clarify or repeat what they said, carrying on the conversation in a quiet place, and withdraw from the conversation. Some participants mentioned that they avoided crowded situations; if they made an exception, they carefully considered their choice: *"That is often the assessment I make when I go to a birthday party: what does it give me and what does it cost me. If there is an equilibrium, it's not so bad, but if there's not, you think: 'I will pass' or 'I will go another time' (male, aged 27)."*

On the other hand, having conversations one-on-one or in a small group (i.e. no more than five people) in a quiet environment, did not cause many difficulties. *"With four people [. . .], I think: 'wait a minute: is the lighting alright? Yes. Can I understand it? It takes some time getting used to the voices. What is the conversation about?' Well, then it is alright (female, aged 51)."* An important factor in comprehending conversations for individuals with DSL was that others did not talk simultaneously and ensured that they spoke clearly. As opposed to busy environments, most participants did benefit from their hearing aids in quiet situations.

**Conversations by telephone.** Participants' experiences regarding telephone conversations varied, however, most participants were able to successfully have a conversation by telephone. Still, some difficulties were mentioned: *"Sometimes it's difficult, because sometimes people talk very fast, or sometimes I cannot understand it for some reason (female, aged 52)."* Other challenges that were mentioned were having difficulties remembering a phone number or not being able to understand everything when background noise is present, which can lead to misunderstandings. Once again, hearing aids were used to make telephone conversations easier: *"When I call my general practitioner I can hear him directly in my ears [via hearing aids], and I can understand him very well (male, aged 46)."* Other strategies regarding telephone use were turning up the volume, explaining to the other person that they were hard of hearing, or only having telephone conversations in a quiet environment.

## Fatigue

Regarding fatigue, participants discussed how they experienced fatigue, what problems it caused, and how they coped with it in their daily lives.

Most participants suffered from fatigue. Some of them experienced fatigue after busy days, others, however, felt fatigued nearly every day: *"It's not something you can skip. It sticks with you. It's not like you get out of bed in the morning and think: 'I'm going to do everything that I want.' You won't make it (male, aged 27)."* Another participant mentioned: *"I try to go to sleep at 10 P.M. [. . .] If I get up at 8 A.M. the next day it's early for me. Often, I still feel tired. [. . .] Some days I think: 'I can never get enough sleep.' But even if my body is well rested, I can still get very tired two hours after I've woken up (female, aged 20)."* Some participants mentioned that their daily energy levels can vary, and that it also depends on the time of day and the time of the year: *"In winter time, [. . .] I don't want to go out in the evening. [. . .] You're already tired in the evening. So when it gets dark at 5 P.M., I have a short amount of time to walk my dog three times, to exercise, to see my friends. [. . .] So in spring and summer, things are going well, [. . .] but when the weather gets worse and it gets darker outside I can do even less (female, aged 42)."* Travelling, communicating in noisy environments, watching television, doing groceries and household tasks were mentioned to be the most tiring activities. These activities were

considered exhausting, because they required considerable concentration on vision and/or hearing: *"When we have meetings at work, I am incredibly tired after a two hour meeting. [. . .] When I continue working at my desk, and someone speaks to me, who I should be able to understand, but I can't. [. . .] I am so tired of all the listening that I cannot understand people any more. [. . .] Then I prefer to continue working visually and not talk anymore, because it requires so much effort to understand it (female, aged 52)."*

Fatigue caused several problems in participants' lives. Some participants experienced physical discomforts, like headaches and eye strains. It also influenced participants' mental wellbeing; fatigue often resulted in frustration, sadness or anger: *"Sometimes I am angry. I think: 'This doesn't make sense at all.' That I just wake up tired, because. . . Yes, because maybe I've done too much in the previous days. And even though I've gone to sleep early, I'm still tired. [. . .] That's just frustrating sometimes, because you don't want to be tired. You just want to be alert (female, aged 20)."* Participants also tended to compare themselves to others: *"You still want to function like healthy people. Then you quickly tend to do the same as everyone else. [. . .] But not everything is possible (male, aged 41)."*

There were several strategies mentioned to handle fatigue. Some participants received advice from low vision services about balancing their energy, which was considered helpful. Others found it helpful to explain their fatigue to others, so they would encounter less incomprehension. Another strategy was to have a regular sleep rhythm. The most frequently mentioned strategy was making a plan for daily activities, and consider which activities are reachable. *"You have to plan very carefully and consider: what are my priorities and what is important and how am I going to divide that over the day. If I have to do three things in a day, I won't say: 'I will do three things in the morning.' I try to divide it over the day with breaks (male, aged 27)."* Participants often felt most energised in the morning, so they started their day with the most important or tiring activity. Participants also mentioned that it was important to not plan too many activities after a busy day, because they often needed the next day to rest. Rest was considered very important, and most participants usually went to bed early. Most participants also rested in the afternoon: *"Not because I have to sleep, but just not hearing any more, not concentrating anymore, just nothing (female, aged 52)."* Others, however, did need sleep during the day: *"I know that if I go to bed now, I'll be asleep soon. But I think it doesn't matter at what time I do it. Okay, I wouldn't do it in the morning. But in the afternoon. . . Nine out of ten times I go lie down in the afternoon, and I'll be asleep for two hours (female, aged 49)."*

## Discussion

This study explored the experiences of individuals with DSL concerning access to information, mobility, communication, and fatigue in general, and how they utilise their vision and hearing in these areas. The results revealed that participants regularly ran into challenges, but also that they were resourceful in finding ways to cope with them. A notable finding was that participants were mostly unaware of how they make use of their residual vision or hearing to compensate, which made the latter part of the research question hard to answer.

The experiences of individuals with DSL found in this study do partly overlap with the experiences of individuals with a single sensory impairment (i.e. a visual or auditory impairment). In the area of access to information, the participants of this study indicated that they often and mostly successfully used multimedia devices. These results are in line with those of a previous study investigating multimedia use of individuals with a visual impairment [28]. Previous studies investigating mobility in individuals with a visual impairment found that they experience challenges in unknown locations [29], a finding that is confirmed by this study. A common problem reported by individuals with hearing loss is that understanding

speech in crowded settings is considered very difficult [30], which is also confirmed by this study. Moreover, some participants mentioned communication breakdown (i.e. being unable to exchange information) in certain situations, a phenomenon also frequently observed in individuals with hearing loss [31]. Fatigue was previously studied in individuals with a visual impairment. Having to put in a lot of effort to visually interpret the environment has been reported as one of the main causes of fatigue [32]. We also found that this was a major cause of fatigue for individuals with DSL.

In addition to this, we identified challenges that were distinct from challenges reported by individuals with a single sensory impairment. For example, in the area of access to information, individuals with DSL often do not enjoy watching television, a finding supported by a previous study investigating multimedia use of individuals with DSL [13]. With regard to mobility, our results suggest that individuals with DSL find it difficult to locate other road users. This was confirmed by another qualitative study investigating independent travel by individuals with DSL [33]. Related to communication, participants with DSL stated difficulties in determining the emotion expressed by the speaker, due to the fact that both facial expressions and voice could not be distinguished. A literature study confirms that difficulties perceiving non-verbal cues can interfere with communication in individuals with DSL [14]. In the area of fatigue, we found that, on top of the effort needed to perceive the visual world, individuals with DSL have to put in additional effort to identify what they hear. These findings support the theory of 1+1 = 3, proposing that challenges related to DSL go beyond solely combining those associated with single sensory impairments.

Although the experienced problems are mostly unique to each area of life, the coping strategies show some overlap. A frequently mentioned coping strategy in all areas of this study was seeking social support. Previous research has demonstrated a positive association between social support and quality of life in other populations [34, 35]. For individuals with DSL, social support has been identified as a protective factor against depression [36]. Two other coping strategies identified in several areas were the use of aids (e.g. hearing aid, cane) and seeking help from low vision services. Similar to seeking social support, the use of cochlear implants and hearing aids in individuals with hearing loss was associated with increased quality of life [37, 38]. Although previous research suggests that many individuals with DSL do not use a hearing aid or cochlear implant (e.g. due to fear of stigmatization) [14], this and other studies indicate that they might benefit from them [39, 40]. A final coping strategy that was present in multiple areas of life was avoidance. Although in general this is not considered an effective coping strategy [41], we found that avoidance was often a well-considered choice. For example, some participants avoided having a conversation in a crowded environment, because it would cost them too much energy. If this is the case, avoidance may be regarded as a reasonable decision. Even though the participants were resourceful in finding coping strategies, the fact that they were in the possession of such a broad range of strategies accentuates the vulnerability of this group.

A strong point of this study is that the qualitative design allowed for a detailed exploration of participants' experiences. Moreover, the mixed sample of participants with regard to gender, age, and severity of DSL contributes to the broadness of the data. A limitation of this study is that nearly half of the interviews were conducted via a video call. In some cases, the internet connections were not stable; as a consequence, these participants' experiences may not have been fully captured. Still, we were able to retrieve valuable information from the audio recordings and the researcher's field notes. Although effort was made to include a diverse sample of participants regarding the severity of vision and hearing loss, individuals with both complete vision and hearing loss were not included in this study. Thus, the findings should be interpreted with care regarding this subgroup. The results might also not be generalizable to the

entire DSL population as a consequence of the small sample size. However, the final interviews provided limited additional information, which implies that data saturation may have been reached. Lastly, due to the fact that most participants found it difficult to indicate how they made use of sensory compensation in several situations, we gathered less data than expected on this subject. Perhaps participants were unaware of sensory compensation, because it is an integrated part of life and has therefore become an automatic process.

In our next study, we will answer the question of how individuals with DSL use sensory compensation. This will be achieved by using quantitative methods, e.g. modeling specific visual and hearing functions in relation to daily activities. The results from this study can also be used to develop interventions aimed specifically at individuals with DSL. Previously, a promising intervention has been developed that addresses, among other things, proper hearing aid use of individuals with DSL [42]. Another recently developed intervention aims to reduce fatigue in individuals with a visual impairment by means of an online training based on cognitive behavioural therapy and self-management [43]. The results of our study may be used to improve and adapt these interventions, and might be of use to enhance existing computer- and mobility trainings given to individuals with DSL in clinical practice.

In clinical practice, individuals with DSL often have to visit separate health care professionals for their vision and hearing problems (e.g. low vision services and audiology services) [3]. Therefore, healthcare professionals' expertise of DSL may be lacking. As this may lead to suboptimal care for individuals with DSL, it is important that professionals are well-informed about possibilities to improve care for individuals with DSL.

As far as we are aware, this is the first qualitative study to investigate the experiences of individuals with DSL in four important areas of life. Through thorough exploration of participants' experiences, we have revealed several challenges and coping strategies encountered by individuals with DSL in access to information, mobility, communication and fatigue. Although some of the results showed overlap with previously conducted research on single sensory impairments, we have also uncovered challenges that are specific to the DSL population, either in nature or in severity. Inventive coping strategies were used by individuals with DSL to handle obstacles encountered in daily life. The results of our study may stimulate the development of DSL-specific interventions and provide healthcare professionals with important information they can use to improve care for individuals with DSL in daily clinical practice.

## Supporting information

**S1 Appendix. Interview guide.**
(PDF)

## Acknowledgments

We are grateful to our study participants for sharing their experiences, and to Ronald Pennings, Carel Hoyng and Stichting Ushersyndroom for letting us use data from their CRUSH study. We also appreciate the efforts of the health care professionals employed by Royal Dutch Visio who helped us develop the interview guide, recruit participants, and retrieve participants' medical information.

## Author Contributions

**Conceptualization:** E. Veenman, A. A. J. Roelofs, A. M. Bootsma, R. M. A. van Nispen.

**Formal analysis:** E. Veenman, M. L. Stolwijk.

**Funding acquisition:** A. A. J. Roelofs, R. M. A. van Nispen.

**Investigation:** E. Veenman.

**Methodology:** E. Veenman, A. A. J. Roelofs, M. L. Stolwijk, A. M. Bootsma.

**Project administration:** A. M. Bootsma.

**Supervision:** A. A. J. Roelofs, R. M. A. van Nispen.

**Visualization:** E. Veenman, A. A. J. Roelofs.

**Writing – original draft:** E. Veenman.

**Writing – review & editing:** A. A. J. Roelofs, M. L. Stolwijk, A. M. Bootsma, R. M. A. van Nispen.

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
