## [Decision Letter · Decision Letter 0]

2 Mar 2023

PONE-D-22-21133Experiences of people with dual sensory loss in various areas of life: A qualitative studyPLOS ONE

Dear Dr. Veenman,

Thank you for submitting your manuscript to PLOS ONE. After careful consideration, we feel that the article has some problems related to methodology. Therefore, we invite you to submit a revised version of the manuscript that addresses the points raised during the review process.

ACADEMIC EDITOR:1. The interview schedule needs to be elaborated. It does not explain its theoretical framework, construction and standardization procedure.2. Describe in detail the data analysis framework in order to justify the conclusions derived. 

We look forward to receiving your revised manuscript.

Kind regards,

Shazia Khalid, PhD

Academic Editor

PLOS ONE

Journal Requirements:

Additional Editor Comments:

You are requested to amend the article as observations of the reviewer at your earliest. Thank you

Reviewers' comments:

Reviewer's Responses to Questions

**Comments to the Author**

1. Is the manuscript technically sound, and do the data support the conclusions?

Reviewer #1: Yes

2. Has the statistical analysis been performed appropriately and rigorously? 

Reviewer #1: Yes

3. Have the authors made all data underlying the findings in their manuscript fully available?

Reviewer #1: No

4. Is the manuscript presented in an intelligible fashion and written in standard English?

Reviewer #1: Yes

5. Review Comments to the Author

Reviewer #1: Thank you for the opportunity to review this paper. It is a very interesting and needed piece of work. Overall well-written and provides great insight into the challenges people with DSL face. My one 'major' comment would be with regard to the method. There are very few studies which are conducted involving people with sensory impairment, particularly people with DSL. Therefore I think beyond just presenting your findings, this can be used as an opportunity to provide more info to other researchers on how to conduct qualitative studies with people with DSL. Therefore adding more details on formats used for participant recruitment and conducting interviews, given that some might not be able to hear well enough, would be good. It is a little unclear if there were those who had both complete vision and hearing impairment in this study. If not, then something about this should be mentioned in the limitations - with regard to generalizability. Similar with those with DSL who use sign language to communicate.

My other comments are attached

6. PLOS authors have the option to publish the peer review history of their article (what does this mean?). If published, this will include your full peer review and any attached files.

Reviewer #1: No

---

## [Author Response · Author response to Decision Letter 0]

22 May 2023

Editorial Board Member’s comments

The interview schedule needs to be elaborated. It does not explain its theoretical framework, construction and standardization procedure.

We have added additional information about the theoretical framework, construction and standardization of the interview guide to the S1 Appendix. We have also added the introduction that was used when conducting the interviews.

“Before constructing this interview guide, we explored literature on dual sensory loss (DSL), and chose four important life areas to focus the interviews on, namely: access to information, mobility, communication and fatigue.

This interview guide was developed in cooperation with healthcare professionals (i.e. computer trainer, mobility trainer, social worker and educationalist) who often encounter clients with DSL and are experts in the areas we were interested in. The interview guide consists of an introduction and open-ended questions about sociodemographic and medical information, and the four areas of interest: access to information, mobility, communication and fatigue. More specifically, the guide consists of questions regarding which challenges or potential favourable factors people experience in these areas, how they cope with these challenges, and how they use their vision and hearing while performing certain activities. The interview was pilot tested under supervision and with a volunteer with DSL.

Introduction

Thank you for your participation in this study. I will first introduce myself and elaborate on the study and the interview itself.

My name is [name] and I am [profession, institution]. In this study, we aim to explore experiences of people with DSL with regard to access to information, mobility, communication, and fatigue, and how they make use of their vision and hearing in these areas. We can then use this information to improve the care provided to people with DSL. To achieve this aim, we will interview people with DSL about how they experience their daily lives regarding these topics. Your experiences and your opinion will be the most important parts of this interview.

Before we start, I would like to discuss some practical matters with you. The interview will last approximately 1 to 1.5 hours. During the interview you can always decline to answer a question. Please also let me know if you do not understand a certain question, if you cannot understand me properly or if you need a break. I also want to emphasize that there are no right or wrong answers. You can also stop the interview at any time, if you wish to do so.

Everything we discuss will be treated confidentially. As described in the information letter, I will make an audio recording of this conversation. The recordings will be transcribed, and they will be destroyed at the end of this study.

I will now discuss the interview itself. I will ask you open questions, and please tell me anything that comes to mind. First of all, we are interested in general information about you and your DSL. Then I will ask you questions about your experiences in the areas of access to information, communication, mobility, and fatigue. We will go through these areas one by one. The questions are similar, yet yield valuable unique information. To ensure that we can discuss all topics, I will keep an eye on the time.

Do you have any questions before we start?”

Describe in detail the data analysis framework in order to justify the conclusions derived.

We elaborated on the method used to analyze the data (framework analysis).

“The data were analysed by means of framework analysis [23] using ATLAS.ti 9 software. To get familiar with the data (step 1), two researchers (EV and MS) read and discussed two interview transcripts. As a second step, the main themes were determined, by combining both a priori knowledge retrieved from the literature and the information gathered from familiarization. This resulted in four main themes which corresponded to the topics of the interview guide (i.e. access to information, mobility, communication and fatigue). Next, two interview transcripts were independently open coded, and by means of face-to-face discussion subthemes were identified. The main themes and subthemes were combined in a codebook, which was used to code two new transcripts. The process of coding and adapting the code book was repeated two times, after which one researcher (EV) coded the remaining transcripts with the final codebook (step 3). During this process, no additional changes were made to the codebook, suggesting that data saturation was reached [24]. Finally, the coded data was summarized for each theme and subtheme (step 4) and interpreted (step 5) by means of comparing the data across participants and identifying patterns in the data.” (Line 142-156)

Reviewer’s comments

Introduction: A brief explanation of Usher syndrome might be good

We have added an explanation of Usher syndrome.

“Wahlqvist et al. [13] investigated the mental health of people with Usher syndrome type 1. This is a rare (3-6 out of 100,000 [14]), genetic disease that is associated with both vision and hearing loss; various types exist, and type 1 is characterised by congenital deafness and progressive vision loss. The authors found that more than sixty percent of the participants had fatigue-related complaints.” (Line 76-80)

Introduction: But why is this info important and to whom?

A sentence which further explains the relevance of the study has been added to the introduction.

“This information may be of importance to healthcare professionals working with the target population as this might help them improve the care they provide, e.g. how clients use their remaining vision and/or hearing as a compensation strategy.” (Line 90-93)

Introduction: This [statement that the aim of the study was partly achieved] should be in the discussion or conclusion

The sentence has been removed. The first paragraph of the discussion elaborates on whether the aim of the study was achieved:

“This study aimed to explore the experiences of people with DSL concerning access to information, mobility, communication, and fatigue in general, and how they utilise their vision and hearing in these areas. The results revealed that participants regularly ran into challenges, but also that they were resourceful in finding ways to cope with them. A notable finding was that participants were mostly unaware of how they make use of their residual vision or hearing to compensate, which made the latter part of the research question hard to answer.” (line 474-480)

Methods: It would be good to incorporate elements of COREQ within the paper e.g. qualifications of researchers etc. Or at the very least this should be appended

More elements of the COREQ checklist were added to the paper. We added the sampling method that we used: “Two channels were used to purposively recruit a sample of participants.” (line 113), and the qualifications of the first author: “Semi-structured, in-depth interviews were conducted by the first author (MSc. Cognitive psychology) (line 126-127). 

Please note that various elements of the COREQ checklist were already incorporated in the paper. For example, information on participant selection: “Two channels were used to purposively recruit a sample of participants. First, health professionals employed by an inpatient low vision service center in the Netherlands (Royal Dutch Visio) were asked to recruit potential participants. […] Second, a call for participation was posted on several social media channels (Facebook, Instagram, Twitter and LinkedIn) of Royal Dutch Visio.” (line 113-118), and data collection: “An interview guide was developed in cooperation with healthcare professionals (e.g computer trainer, mobility trainer) who often encounter clients with DSL and are experts in the areas of interest. The interview guide consisted of an introduction and open-ended questions about sociodemographic and medical information, and the four areas of interest mentioned earlier (see Appendix S1). […] Interviews were audio-recorded and transcribed verbatim.” (line 127-136).

Methods: Were DSL patients who were Deaf i.e. used sign language, included? Or was that an exclusion?

Deaf participants were not excluded from this study. We interviewed two participants who made use of sign language. A speech-to-text interpreter was present to facilitate the interview. This comment made us notice a mistake in our manuscript. In line 110-111, it was stated that being dependent on alternative communication (e.g. sign language) was an exclusion criterion. This is incorrect, thus we removed this phrase.

Methods: Why was it limited to this center? Why not also include one for hearing impairment? Did they not have people with DSL?

Royal Dutch Visio initiated this study and was our starting point for recruitment, and we managed to include the amount of participants that we strived for through the two channels that we mentioned in the manuscript. For the recruitment of participants in future studies, we indeed plan to involve centers who provide care for people with a hearing impairment.

Methods: Was the guide validated and pilot tested?

Yes, interview questions were developed in cooperation with healthcare professionals who have experience with clients with dual sensory loss (DSL). The professionals that were involved were a computer trainer, a mobility trainer, a social worker and an educationalist. 

The guide was also pilot tested; a former client of Royal Dutch Visio with DSL volunteered to practice the interview with EV. RvN (supervisor/professor, background in clinical psychology) was also present, and both provided feedback afterwards. Minor changes to the interview guide were made as a result of this pilot interview. The data gathered during this interview was also used in the final data set. We added to the manuscript that: “the interview was pilot tested under supervision and with a volunteer with DSL.” (line 134-135)

Methods: I find this [i.e. that the study is not subject to the Medical Research Involving Human Subjects Act (Dutch law)] quite shocking especially given the population involved.

The Dutch Medical Research Involving Human Subjects Act is only applicable to certain studies (e.g. pharmaceutical studies, randomized controlled trials); this act typically does not apply to qualitative studies. Of course, other laws do apply to this study (e.g. privacy protection laws).

Results: no need to state gender in the table since already mentioned in the text

In our opinion, it is important that the information on gender can be easily found while scanning the paper, thus we chose to leave it in the table and removed it from the text instead (line 165). We hope the editor and reviewer agree with this suggestion.

Results: State this in the methods including amount

This sentence has been moved to the Methods, and the amount of the gift voucher has been added: 

“Participants received a €20 gift voucher after finishing the interview.” (Line 135).

Results: There should be a paragraph before this to outline the themes and subthemes that were developed.

A paragraph is added to outline the main themes and subthemes.

“Four main themes were derived from the data: access to information, mobility, communication and fatigue. Three of these main themes were divided in subthemes; for access to information, these were smartphone, computer and tablet usage and watching television. Additionally, mobility and communication were divided into subthemes of walking, the use of public transportation, face to face conversation and conversations by telephone, respectively.” (Line 188-193)

Discussion: Can you elaborate a bit on these interventions

A brief explanation of the communication intervention for people with DSL and the fatigue intervention for people with a visual impairment has been added.

“Previously, a promising intervention has been developed that addresses, among other things, proper hearing aid use of people with DSL [39]. Another recently developed intervention aims to reduce fatigue in people with a visual impairment by means of an online training based on cognitive behavioral therapy and self-management [40].” (Line 550-554)

Discussion: This is too brief - how else can this information be used? Thinking about the gaps faced and what is needed. There should be a separate section on recommendation for future research and how the info from this study can be used. Conclusion should be a bit longer and conclude the study better

We elaborated a bit more on the implications of this study and recommendations for future research in a separate section. 

“In our next study, we will answer the question of how people with DSL use sensory compensation. This will be achieved by using quantitative methods, e.g. modeling specific visual and hearing functions in relation to daily activities. The results from this study can also be used to develop interventions aimed specifically at people with DSL. Previously, a promising intervention has been developed that addresses, among other things, proper hearing aid use of people with DSL [39]. Another recently developed intervention aims to reduce fatigue in people with a visual impairment by means of an online training based on cognitive behavioural therapy and self-management [40]. The results of our study may be used to improve and adapt these interventions, and might be of use to enhance existing computer- and mobility trainings given to people with DSL in clinical practice.” (Line 546-556)

We agree that the conclusion can be posed a little stronger:

“As far as we are aware, this is the first qualitative study to investigate the experiences of people with DSL in four important areas of life. Through thorough exploration of participants’ experiences, we have revealed several challenges and coping strategies encountered by people with DSL in access to information, mobility, communication and fatigue. Although some of the results showed overlap with previously conducted research on single sensory impairment, we have also uncovered challenges that are unique to the DSL population. Inventive coping strategies were used to make life with DSL as enjoyable as possible. The results of our study may stimulate the development of DSL-specific interventions and provide healthcare professionals with important information they can use to improve care of people with DSL in daily clinical practice.” (Line 568-578)

Thank you for the opportunity to review this paper. It is a very interesting and needed piece of work. Overall well-written and provides great insight into the challenges people with DSL face. My one 'major' comment would be with regard to the method. There are very few studies which are conducted involving people with sensory impairment, particularly people with DSL. Therefore I think beyond just presenting your findings, this can be used as an opportunity to provide more info to other researchers on how to conduct qualitative studies with people with DSL. Therefore adding more details on formats used for participant recruitment and conducting interviews, given that some might not be able to hear well enough, would be good. It is a little unclear if there were those who had both complete vision and hearing impairment in this study. If not, then something about this should be mentioned in the limitations - with regard to generalizability. Similar with those with DSL who use sign language to communicate.

We thank the reviewer for the kind words about our paper. We agree that the paper can be a bit more elaborate with regard to participant recruitment and conducting the interviews. We have made some adjustments to clarify this. Additionally, two of our participants used sign language to communicate; we also clarified this in the text.

“If people were interested to participate, they contacted (i.e. telephoned or emailed) the executive researcher, who provided further information, and sent them an information letter and informed consent form (either a physical or digital copy, depending on the participants’ preferences).” (Line 118-121)

“Most participants were capable of having a spoken conversation. However, two participants used sign language as their main means of communication, thus, a speech-to-text interpreter was present at the interview. The interpreter transformed the spoken words of the interviewer to text, which participants could read, in a large type font, on a computer screen. These two participants provided their answers in spoken form.” (Line 179-184)

In the discussion, we added the suggested limitation on the absence of participants with both complete vision and hearing loss. 

“Although effort was made to include a diverse sample of participants regarding the severity of vision and hearing loss, people with both complete vision and hearing loss were not included in this study. Thus, the findings should be interpreted with care regarding this subgroup.” (Line 535-538)

Journal requirements

After reviewing the PLOS ONE’s style requirements, we believe that our manuscript meets these.

We note that you have indicated that data from this study are available upon request. PLOS only allows data to be available upon request if there are legal or ethical restrictions on sharing data publicly. 

If there are ethical or legal restrictions on sharing a de-identified data set, please explain them in detail (e.g., data contain potentially sensitive information, data are owned by a third-party organization, etc.) and who has imposed them (e.g., an ethics committee). Please also provide contact information for a data access committee, ethics committee, or other institutional body to which data requests may be sent.

The qualitative data from the interviews contain potentially sensitive information and cannot be de-identified. Dutch privacy law states that these data cannot be made public. A summary of the data as well as the code book used to analyze the data, was posted to an online repository (https://doi.org/10.17026/dans-zpf-57wx).

Additional sharing conditions will be considered while respecting the type of informed consent provided by the subjects and discussed with the support of RDM experts where appropriate.

We reviewed the reference list and believe it is complete and correct.

---

## [Decision Letter · Decision Letter 1]

12 Jul 2023

PONE-D-22-21133R1

Experiences of people with dual sensory loss in various areas of life: A qualitative study

PLOS ONE

Dear Dr. Veenman,

Thank you for submitting your manuscript to PLOS ONE. After careful consideration, we feel that it has merit but does not fully meet PLOS ONE’s publication criteria as it currently stands. Therefore, we invite you to submit a revised version of the manuscript that addresses the points raised during the review process.

We look forward to receiving your revised manuscript.

Kind regards,

Shazia Khalid, PhD

Academic Editor

PLOS ONE

Journal Requirements:

Reviewers' comments:

Reviewer's Responses to Questions

**Comments to the Author**

1. If the authors have adequately addressed your comments raised in a previous round of review and you feel that this manuscript is now acceptable for publication, you may indicate that here to bypass the “Comments to the Author” section, enter your conflict of interest statement in the “Confidential to Editor” section, and submit your "Accept" recommendation.

Reviewer #2: All comments have been addressed

2. Is the manuscript technically sound, and do the data support the conclusions?

Reviewer #2: Partly

3. Has the statistical analysis been performed appropriately and rigorously? 

Reviewer #2: N/A

4. Have the authors made all data underlying the findings in their manuscript fully available?

Reviewer #2: No

5. Is the manuscript presented in an intelligible fashion and written in standard English?

Reviewer #2: No

6. Review Comments to the Author

Reviewer #2: The paper titled "Experiences of People with Dual Sensory Loss in Various Areas of Life: A Qualitative Study" presents a qualitative exploration of the lived experiences of individuals who have dual sensory loss. Its a good addition in the filed, however, following suggestions may enhance its quality of manuscript.

1. The manuscript would greatly benefit from a thorough proofreading. There are several instances where grammatical errors, typos, or inconsistencies in punctuation were observed. A careful review of the document is necessary to ensure clarity and readability. Line no. 100, 103, 122, 167, 172, are a few examples for reference

2. This paper explores the four domains of access to information, mobility, communication, and fatigue. However, the rationale behind the selection of these specific domains is not explicitly stated. It is imperative to thoroughly examine and articulate the significance of addressing these domains in order to enhance the understanding and relevance of the subject matter.

3. In the introductory chapter, the author presents their findings, which may not be deemed suitable for inclusion in this section. Such findings are typically discussed in the appropriate chapter, such as the discussion chapter. For instance, specific lines, such as 73 and 79, may need to be relocated to the relevant sections where their context and implications can be properly addressed.

4. Although the manuscript provides literature support, it is observed that there is a lack of theoretical underpinning in the presented work.

5. In lines 85-86, the author employs the term "aforementioned" instead of using the appropriate and precise wording, which results in an inaccurate narrative. It is important to adhere to the guidelines outlined by the APA style, where authors are expected to use the actual words rather than relying on such general references.

6. In lines 90-93, the author engages in a discussion regarding the implications of the study, while it is expected that they should focus on articulating the significance of the problem being addressed and emphasizing the importance of providing a viable solution. It is essential to direct attention towards highlighting the relevance and value of both the problem and its corresponding solution within the given context.

7. PLOS authors have the option to publish the peer review history of their article (what does this mean?). If published, this will include your full peer review and any attached files.

Reviewer #2: No

---

## [Author Response · Author response to Decision Letter 1]

24 Aug 2023

The manuscript would greatly benefit from a thorough proofreading. There are several instances where grammatical errors, typos, or inconsistencies in punctuation were observed. A careful review of the document is necessary to ensure clarity and readability. Line no. 100, 103, 122, 167, 172, are a few examples for reference

We asked one of our research fellows who is a native speaker to check the manuscript and we corrected some issues throughout the entire manuscript. A few examples are given below.

“In-depth interviews were conducted with Dutch-speaking adults, with varying severities of both vision and hearing loss.” has been changed to “Comprehensive interviews were undertaken with Dutch-speaking adults, who exhibited diverse degrees of impairment in both vision and hearing.” (Line 121-122)

“In total, 20 interviews were planned after which data saturation would be expected.” has been changed to “In total, 20 interviews were scheduled which we anticipated to be the number required to achieve data saturation.” (Line 141-143)

“In 70% of the participants binaural hearing aids were used. In Fig 1, the participants’ visual acuity, visual field loss and hearing loss is presented.” has been changed to “Binaural hearing aids were utilized by 70% of the participants. Fig 1 shows the participants’ visual acuity, visual field and hearing loss information of all 20 participants included in this study.” (Line 188-191)

This paper explores the four domains of access to information, mobility, communication, and fatigue. However, the rationale behind the selection of these specific domains is not explicitly stated. It is imperative to thoroughly examine and articulate the significance of addressing these domains in order to enhance the understanding and relevance of the subject matter.

We agree that the rationale can be stated more clearly. We added the following sentences to the introduction:

“Deafblind International and both the Nordic and Dutch definitions of deafblindness mention access to information, mobility and communication as life areas that may be problematic for people with DSL [10-12].” (Line 70-73)

“The Dutch functional definition of deafblindness identifies fatigue as another important life area that can cause challenges for individuals with DSL.” (Line 84-85)

In the introductory chapter, the author presents their findings, which may not be deemed suitable for inclusion in this section. Such findings are typically discussed in the appropriate chapter, such as the discussion chapter. For instance, specific lines, such as 73 and 79, may need to be relocated to the relevant sections where their context and implications can be properly addressed.

The examples mentioned by the reviewer refer to previous research as opposed to the current study. We adjusted the wording to make this clearer.

“Additionally, a review of DSL literature confirms that problems in the areas of communication and mobility are often experienced by individuals with DSL [14]. For example, it was found that difficulties in identifying facial expressions negatively influenced speech understanding.” (Line 80-83)

“The results of this study showed that more than sixty percent of the participants had fatigue-related complaints.” (Line 91-92)

Although the manuscript provides literature support, it is observed that there is a lack of theoretical underpinning in the presented work.

There has been relatively little research with regard to dual sensory loss. As a consequence, presented theories are limited. 1+1=3 can be regarded as theoretical underpinning for this study, further explained in the paper:

“However, having both visual and auditory impairments makes the compensatory function of both senses less obvious, and seems to result in difficulties that may go beyond the consequences of having a single sensory impairment; a principle that can be exemplified by 1+1=3 [1, 9].” (Line 65-68)

In the discussion, we refer back to 1+1=3, in the section highlighting the challenges specific to DSL.

“In addition to this, we identified challenges that were distinct from challenges reported by individuals with a single sensory impairment. For example, in the area of access to information, individuals with DSL often do not enjoy watching television, a finding supported by a previous study investigating multimedia use of individuals with DSL [13]. With regard to mobility, our results suggest that individuals with DSL find it difficult to locate other road users. This was confirmed by another qualitative study investigating independent travel by individuals with DSL [33]. Related to communication, participants with DSL stated difficulties in determining the emotion expressed by the speaker, due to the fact that both facial expressions and voice could not be distinguished. A literature study confirms that difficulties perceiving non-verbal cues can interfere with communication in individuals with DSL [14]. In the area of fatigue, we found that, on top of the effort needed to perceive the visual world, individuals with DSL have to put in additional effort to identify what they hear. These findings support the theory of 1+1=3, proposing that challenges related to DSL go beyond solely combining those associated with single sensory impairments.” (Line 523-539)

In lines 85-86, the author employs the term "aforementioned" instead of using the appropriate and precise wording, which results in an inaccurate narrative. It is important to adhere to the guidelines outlined by the APA style, where authors are expected to use the actual words rather than relying on such general references.

We agree that this sentence is confusing. We added the life areas that we meant to refer to.

“Previous qualitative research on DSL did focus on one or more of the aforementioned areas (i.e. access to information, mobility, communication, and fatigue) [19, 21], but were often aimed at specific groups (e.g. older adults [19] or individuals with Usher syndrome [21]).” (Line 97-100)

In lines 90-93, the author engages in a discussion regarding the implications of the study, while it is expected that they should focus on articulating the significance of the problem being addressed and emphasizing the importance of providing a viable solution. It is essential to direct attention towards highlighting the relevance and value of both the problem and its corresponding solution within the given context.

We tried to give more emphasis to the problem in this section.

“A smaller number of qualitative studies have focused on compensation strategies of individuals with DSL [22], however, to our knowledge, no qualitative studies have been conducted in which the role of vision and hearing as compensatory senses in DSL was addressed. This information might allow individuals with DSL to benefit from strategies (e.g. learn how to use their remaining vision and/or hearing as a compensation strategy) that lead to improved access to information, mobility, communication and to less fatigue. Consequently, this information may also be of importance to healthcare professionals working with the target population as this might help them improve the care they provide.” (Line 100-108)

---

## [Editor Report · Decision Letter 2]

29 Aug 2023

Experiences of people with dual sensory loss in various areas of life: A qualitative study

PONE-D-22-21133R2

Dear Author,

We’re pleased to inform you that your manuscript has been judged scientifically suitable for publication and will be formally accepted for publication once it meets all outstanding technical requirements.

You’ll receive a formal acceptance letter and your manuscript will be scheduled for publication.

Kind regards,

Shazia Khalid, PhD

Academic Editor

PLOS ONE

---

## [Editor Report · Acceptance letter]

1 Sep 2023

PONE-D-22-21133R2 

Experiences of people with dual sensory loss in various areas of life: A qualitative study 

Dear Dr. Veenman:

I'm pleased to inform you that your manuscript has been deemed suitable for publication in PLOS ONE. Congratulations! Your manuscript is now with our production department. 

Kind regards, 

on behalf of

Professor Shazia Khalid 

Academic Editor

PLOS ONE